

# Goal conflict in chronic pain: day reconstruction method

Nathalie Claes[1,2], Johan W.S. Vlaeyen[1,3], Emelien Lauwerier[2,4], Michel Meulders[5,6] and Geert Crombez[2,7]

[1] Research Group Health Psychology, KU Leuven, Leuven, Belgium
[2] Department of Experimental-Clinical and Health Psychology, Ghent University, Ghent, Belgium
[3] Department of Clinical Psychological Science, University of Maastricht, Maastricht, the Netherlands
[4] Department of Public Health and Primary Care, Ghent University, Ghent, Belgium
[5] Center for Information Management, Modeling and Simulation, KU Leuven, Leuven, Brussels, Belgium
[6] Research Group on Quantitative Psychology and Individual Differences, KU Leuven, Leuven, Belgium
[7] Centre for Pain Research, University of Bath, Bath, UK

## ABSTRACT

**Background:** When suffering from chronic pain, attempts to control or avoid pain often compete with other daily activities. Engaging in one activity excludes engaging in another, equally valued activity, which is referred to as "goal conflict." As yet, the presence and effects of goal conflicts in patients with chronic pain remain poorly understood.

**Methods:** This study systematically mapped the presence and experience of goal conflicts in patients with fibromyalgia compared to healthy controls. A total of 40 patients and 37 controls completed a semi-structured interview in which they first reconstructed the previous day, identified conflicts experienced during that day, and classified each of the conflicting goals in one of nine goal categories. Additionally, they assessed how they experienced the previous day and the reported conflicts.

**Results:** Results showed that patients did not experience more goal conflicts than healthy controls, but that they did differ in the type of conflicts experienced. Compared to controls, patients reported more conflicts related to pain, and fewer conflicts involving work-related, social or pleasure-related goals. Moreover, patients experienced conflicts as more aversive and more difficult to resolve than control participants.

**Discussion:** This study provides more insight in the dynamics of goal conflict in daily life, and indicates that patients experience conflict as more aversive than controls, and that conflict between pain control (and avoidance) and other valued activities is part of the life of patients.

# INTRODUCTION

The fear-avoidance model of chronic pain (*Vlaeyen & Linton, 2000*, *2012*) essentially describes two possible cognitive-behavioral responses to pain. On the one hand, the

Corresponding author
Nathalie Claes,
nathalie.claes@kuleuven.be

individual may appraise pain as nonthreatening, and gradually resumes activities. On the other hand, pain may be interpreted as a sign of injury, which in turn may lead to pain-related fear, resulting in avoidance behavior and (hyper) vigilance. When such pattern of avoidance persists, it may bring along depression, social isolation, disability or reduced participation in daily life activities. Although there is evidence validating these behavioral responses (*Leeuw et al., 2007*; *Zale et al., 2013*; *Wertli et al., 2014*), challenges remain (*Crombez et al., 2012*).

There is a call for including a broad motivational context into the model: patients with chronic pain often not only want to avoid pain, but may also want to pursue other valued activities, such as socializing with friends (*Crombez et al., 2012*; *Vlaeyen, Crombez & Linton, 2009*). Different relations may exist between pain avoidance goals and other goals. Avoiding pain may facilitate pursuing other activities ("goal facilitation"), but it may also interfere with goals ("goal interference"; *Boudreaux & Ozer, 2012*; *Riediger & Freund, 2004*). We may expect that goal interference is often preceded by the experience of goal conflict. Indeed, goal conflicts arise because of incompatible attainment strategies or resource constraints (e.g., time) and is characterized by a behavioral indecisiveness (*Lewin, 1935*; *Miller, 1944*; *Riediger & Freund, 2004*). The responses described by the fear-avoidance model can be reframed in motivational terms: the pattern of avoidance may correspond with prioritizing the goal to control pain at the cost of other goals, whereas the confrontational response may reflect prioritizing and engaging in other life goals, despite pain (*Crombez et al., 2012*; *Lauwerier et al., 2012*; *Van Damme, Crombez & Eccleston, 2008*; *Vlaeyen, Morley & Crombez, 2016*). Although there is research on avoidance and confrontation, there is almost no research on goal conflict. In general, research has demonstrated that experiencing goal conflict negatively affects well-being (*Boudreaux & Ozer, 2012*; *Emmons & King, 1988*). *Karoly et al. (2008)* also reported that patients experience more goal frustration and more goal conflict than control participants. Furthermore, goal conflict has been associated with more pain-related fear (*Karoly et al., 2008*), and with a greater increase in pain from morning to evening (*Hardy, Crofford & Segerstrom, 2011*). However, the potentially detrimental effects of goal conflict on well-being have not always been replicated (*Segerstrom & Solberg Nes, 2006*), suggesting that contextual factors may play a role (*Gorges, Esdar & Wild, 2014*).

Here, we seek to further our understanding of goal conflict in patients with chronic pain. The main objective was exploratory in nature, and focuses on mapping the presence and experience of goal conflicts in patients with fibromyalgia and in healthy controls. Research questions were (1) do patients experience more goal conflict in daily life than healthy participants; (2) do patient and healthy participants differ in the type of conflicts they experience; (3) which goals commonly compete with pain-related goals; (4) do patients and controls differ in the experience and context of conflict; and (5) can core constructs of the fear-avoidance model or individual differences predict the number of (pain-related) goal conflicts?

To this purpose, patients with fibromyalgia and matched healthy controls were invited to participate in a semi-structured interview based on the day reconstruction method (DRM) (*Kahneman et al., 2004*) in which patients first reconstructed the previous day in

keywords. Next, participants identified conflicts experienced during the previous day. Subsequently, participants classified each goal of their conflict in one of the pre-defined categories. Finally, participants assessed the experience of maximally three conflicts and rated their pain, fatigue, emotions, and overall experience of that day. Participants also completed a series of questionnaires.

## MATERIALS AND METHODS

### Participants

The current study is part of the Pain-Attention-Motivation Project 1 (PAM-I-Project; *Claes et al., 2015*), consisting of three independent studies investigating attentional and motivational processes in patients with chronic pain. For an overview of the project, the participant inclusion process and overview of the measurements, see *Claes et al. (2015)*. The PAM-I-Project was approved by the Medical Ethical Committee of Ghent University Hospital (registration number B670201421583). All participants received reimbursement for their expenses.

#### *Patients with fibromyalgia*

Patients with fibromyalgia seeking health care between the ages of 18 and 65 years were recruited in two ways: (a) From July 2011 to August 2014, posters were placed in the waiting room of the Multidisciplinary Pain Centre of Ghent University Hospital, and medical doctors informed patients about the possibility to participate in research. Eighty-four interested patients with fibromyalgia provided their information to be contacted for participation; (b) From August 2014 onwards, patients from the Multidisciplinary Pain Centre are asked to complete online questionnaires at intake. Upon completion of these questionnaires, participants provide their contact details for research purposes. Fourteen individuals with fibromyalgia left their contact information. In sum, both recruitment methods led to a total number of 98 individuals with fibromyalgia who could be contacted. Inclusion criteria were: being diagnosed with Fibromyalgia, fluency in the Dutch language, normal or corrected-to-normal eyesight, normal or corrected-to-normal hearing. Participants were excluded if they suffered from neurological problems (e.g., epilepsy), or reduced tactile sensitivity as this was relevant for another, but unrelated study of the PAM-I-project.

We contacted 90 (91.8%) of the 98 candidates until the predetermined number of 40 participants was reached. Fifty (51%) of the 90 patients did not wish to participate. Most common reasons for non-participation were distance to the faculty, time constraints, or aggravation of complaints. In total, 40 patients with fibromyalgia (three males) participated. Patients were between 29 and 64 years of age ($M = 45.8$, SD = 9.22). The majority of patients was married (57.5%), or cohabiting (5%). Fifteen (37.5%) patients received higher education. Only 22.5% of patients was employed, 5% was retired, and 7.5% was unemployed. The remaining patients received health insurance (17.5%) or disability (47.5%) benefits. The mean reported duration of patients' pain was $14.5 \pm 12.01$ years.

### Healthy control participants

We recruited control participants matching sex, age, and educational level of the fibromyalgia patients via frequency sampling. Healthy participants were recruited in several ways: advertisements in local newspapers or social media, flyers distributed around the university campus, and public venues. A total of 181 candidate individuals expressed their willingness to participate in research and left their contact information. We contacted control participants based on the recruitment of patients: we randomly contacted a candidate control participant that matched for sex, age, and educational level of the patient participants until we found enough candidates willing to participate. We contacted 55 (30.39%) of these 181 candidates; 126 (69.61%) of candidates were not contacted, as they did not match the participant profile (age, sex, educational level) or a sufficient number of control participants was reached. Fourteen out of 55 (23.6%) did not wish to participate. Most common reasons for non-participation were suffering from a chronic illness and lack of time. In total, 41 controls participated. Inclusion and exclusion criteria were the similar, except for the following: fulfilling the criteria for fibromyalgia of the American College of Rheumatology (ACR criteria) for fibromyalgia (*Wolfe et al., 2010*), and suffering from pain of a severe intensity (category II, III or IV, see further) according to the criteria of *Von Korff et al. (1992)*. Three participants suffered from pain of a severe intensity, another met the diagnostic criteria for fibromyalgia. These four (1%) participants were excluded from analyses. The final sample comprised of 37 healthy controls (four males), with a mean age of $45.92 \pm 10.14$ years. Most control participants were either married (29.7%) or living together with a partner (16.2%). A total of 40.5% finished higher education. The majority of control participants was in paid employment or received education (62.2%), 5.4% was retired, and 27% was unemployed. One participant was in unpaid employment, and another received health insurance benefits.

Control participants were matched to patient participants, as they did not significantly differ from patient participants in terms of gender, $t(75) < 1$, $p = 0.619$, age, $t(75) < 1$, $p = 0.957$, level of education, $t(75) = -1.31$, $p = 0.194$, and in marital status, $t(75) < 1$; $p = 0.419$. However, patients were more often unemployed or receiving disability benefits than control participants, $t(75) = -6.775$, $p < 0.0001$.

All participants provided verbal and written informed consent and were informed that participation was voluntary and could be stopped at any point in time, without negative consequences.

## Procedure

Participants were invited for an individual appointment at Ghent University, which took approximately 3 h. Before the individual appointment, participants were asked to complete a sociodemographical information sheet (i.e., age, gender, profession, education level, work status) and several questionnaires. Patients additionally provided information on their pain problem, and completed questionnaires (for an overview of all questionnaires, see the PAM-I-Protocol). Seventy participants filled in these questionnaires online, seven participants filled in a paper version. Questionnaires were included either for descriptive purposes (e.g., sociodemographical information; pain severity), assessing

inclusion and exclusion criteria (e.g., diagnostic criteria for Fibromyalgia; pain severity), and/or exploring the predictive value of the constructs (for example, Depression Anxiety and Stress Scales (DASS), pain catastrophizing scale (PCS), Experience of Cognitive Intrusion Pain scale (ECIP)) in the experience of goal conflict. As this study was part of a large project, a number of questionnaires were not included in the analysis of this study.

During the individual appointment, participants completed a semi-structured interview based on the DRM (*Kahneman et al., 2004*). This semi-structured interview was constructed by a group of (pain research) experts, and was extensively piloted in patients prior to the study. Interviewers (N.C., N.D., E.D.M., J.M; all female) were extensively trained in using the standardized interview protocol. During the interview, participants reconstructed their previous day, next reported the number of goal conflicts experienced during that day, categorized the goals involved, and assessed the emotions and overall experience of the conflict(s). Lastly, participants assessed their pain, fatigue, emotions, and general experience of that day. The interview lasted about 60–90 min per participant.

## Materials and measures
### Sociodemographic information
For descriptive purposes, participants provided information on gender, age, education, employment, and marital status. Patients also provided information on the duration and treatment of their pain problem.

### Diagnostic criteria for fibromyalgia
Participants completed the Dutch version of the ACR Criteria for fibromyalgia (*Geenen & Jacobs, 2010*; *Wolfe et al., 2010*), which consists of two parts. In the first part, respondents indicate the painful locations on a manikin. A widespread pain index is calculated by counting the number of reported painful body regions. The score varies between 0 and 19. In the second part, respondents report on the severity of their cognitive symptoms and the presence of extra somatic symptoms (e.g., headache, fever, tinnitus) using a four-point scale (0 = absent; 3 = a lot). The sum of these items results in a symptom score, ranging from 0 to 12.

### Pain severity
To assess pain severity, the Graded Chronic Pain Scale (GCPS; *Von Korff et al., 1992*) was completed. The GCPS was used to address the exclusion criteria for control participants. Items measuring pain intensity are: current pain intensity, worst pain intensity, and average pain intensity in the past 6 months, using an 11-point scale (0 = no pain; 10 = pain as bad as could be). Items measuring pain disability are: the number of days that the participant was unable to perform his/her usual activities (work, school, or housework) during the past 6 months, the extent of interference with daily activities, the ability to take part in recreational, social and family activities, and the ability to work. The latter three items are scored using an 11-point scale (0 = no interference; 10 = unable to carry on any activities). Based on the pain intensity and interference, respondents can be classified in five categories: (1) Grade 0: no pain in the past 6 months; (2) Grade I: low pain

intensity and low disability; (3) Grade II: high pain intensity, but low disability; (4) Grade III: highly disabling, moderately limiting pain; (5) Grade IV: highly disabling, severely limiting pain. The GCPS has been shown to be a valid and reliable instrument (*Von Korff et al., 1992*).

### Pain catastrophizing

To measure the frequency of catastrophic thoughts and feelings experienced when in pain, participants completed the Dutch version of the PCS-DV (*Crombez et al., 1998*; *Sullivan, Bishop & Pivik, 1995*). The PCS comprises of 13 items, and is scored using a 5-point scale (0 = not at all; 4 = always). The PCS yields a total score between 0 and 52, and three subscale scores: rumination (e.g., "*I keep thinking about how much it hurts*"), magnification (e.g., "*I become afraid that the pain will get worse*"), and helplessness (e.g., "*I feel I can't go on*"). Internal consistency and validity of the PCS are shown to be good (*Sullivan, Bishop & Pivik, 1995*; *Van Damme et al., 2002*). Cronbach's α for the PCS in this study was 0.94.

### Depression, anxiety, and stress

Participants filled in the DASS (*Lovibond & Lovibond, 1995a*, *1995b*), which consists of 42 items describing negative symptoms. Respondents are asked to rate the extent to which they have experienced each of the symptoms during the past week using a four-point numerical scale (0 = not at all applicable; 3 = definitely applicable). Scores for the depression, anxiety, and stress subscales are calculated by summing the corresponding items (14 per subscale). Example items are "*I felt I was pretty worthless*" for depression, "*I felt terrified*" for anxiety, and "*I found that I was very irritable*" for stress. Internal consistency and validity of the DASS are good (*Antony et al., 1998*). In this study, we found a Cronbach's α of 0.94 for stress, 0.89 for Anxiety, and 0.95 for Depression.

### Trait anxiety

To measure trait anxiety, the Dutch translation of the trait version of the Spielberger State-Trait Anxiety Inventory (STAI; *Spielberger, Gorsuch & Lushene, 1970*), called the Zelf-Beoordelings Vragenlijst (*Van Der Ploeg, 1980*), was completed. The STAI trait version consists of 20 items, each rated on a four-point numerical scale (1 = no anxiety; 4 = very anxious). The total score ranges between 20 and 80, with scores of 50 or above labeled as anxious. The STAI has shown to be valid and reliable (*Spielberger, Gorsuch & Lushene, 1970*; *Van Der Ploeg, 1980*). Cronbach's α for this study was 0.94.

### Cognitive intrusions

The ECIP was used to measure the extent to which the experience of pain interferes with thinking when experiencing pain (*Attridge et al., 2015*). The scale has 10 items, all scored on a seven-point scale (0 = not at all applicable; 6 = highly applicable). Items focus on interruption by pain (e.g., "*pain interrupts my thinking*"), ruminative thoughts on pain (e.g., "*pain goes around and around in my head*"), and control by pain (e.g., "*I can't push pain out of my thoughts*"). The total score ranges from 0 to 60, and is obtained by summing all items. Cronbach's α for the ECIP in this study was 0.97.

### Positive and negative affectivity

Participants completed a Dutch version of the trait version of the Positive and Negative Affectivity Scale (PANAS; *Engelen et al., 2006*; *Watson, Clark & Tellegen, 1988*). The PANAS consists of 20 items, 10 positive affect words (e.g., *interested, cheerful*), and 10 negative affect words (e.g., *sad, guilty*). Respondents used a five-point Likert scale (1 = very slightly or not at all; 5 = extremely) to indicate the extent to which they generally experience each of the emotions. This Dutch version of the PANAS is shown to be a reliable and valid instrument (*Engelen et al., 2006*). The Cronbach's α was 0.87 for the positive scale, and 0.90 for the negative scale.

### Pain disability

To measure the degree to which pain interferes with the ability to participate in daily life, we used the pain disability index (PDI; *Pollard, 1984*). This questionnaire consists of seven items assessing the disability in each of the following domains: *family and home responsibilities*, *recreation*, *social activity*, *occupation*, *sexual behavior*, *self-care*, and *life-supporting activity* (e.g., eating) using an eleven point numerical scale (0 = no disability, 10 = total disability). The PDI is considered a reliable and valid instrument to study pain-related disability (*Tait, Chibnall & Krause, 1990*). In the current study, we found a Cronbach's α of 0.87 for the PDI.

### Vigilance

Patient participants completed the Dutch version of the Pain Vigilance and Awareness Questionnaire (PVAQ), which contains 16 items that measure the respondent's vigilance for painful sensations during the last 2 weeks (*McCracken, 1997*; *Roelofs et al., 2002*). Each item is rated on a six-point numerical scale (0 = never; 5 = always). The total score is calculated by summing all items, resulting in a total score ranging from 0 to 80. The validity and reliability of the PVAQ has shown to be good (*Roelofs et al., 2002*, *2003*). Cronbach's α in this study was 0.87.

### Pain-related fear

To assess four components—fearful appraisal of pain, cognitive anxiety, psychological anxiety, and escape and avoidance behavior—of pain-related fear, patient participants completed the Pain Anxiety Symptoms Scale (PASS; *McCracken, Zayfert & Gross, 1992*). The PASS contains 40 items scored on a six-point scale ranging from 0 ("never") to 5 ("always"). The PASS has been shown to be reliable (*Burns et al., 2000*; *Roelofs et al., 2004*). For the PASS, we found a Cronbach's α of 0.86.

### Semi-structured interview

Participants completed a semi-structured interview based on the DRM of *Kahneman et al. (2004)*, which was originally developed to study activities and affective experiences of the previous day. The semi-structured interview used here had the goal to activate memories of the previous day by letting participants reconstruct their day, and to enable them to identify and report on experiences of goal conflict.

*Reconstruction of previous day*

First the interviewer explained the objective and procedure of the interview to participants. Participants indicated the date and day of the previous day, as well as the time they woke up in the morning and the time they went to bed. In contrast with the original DRM—where participants independently reconstruct their previous day by means of an anonymous diary—the interviewer asked participants to verbally report on the activities they had undertaken the previous day. The interviewer prompted participants to freely report the activities of the previous day, and to take the time needed to reflect on that day and on possible key words describing these activities. Participants were asked to report on activities during the morning (from waking until noon), afternoon (noon until about 18:00), and evening (from about 18:00 until going to bed). An activity usually varied between 15 min and 2 h, and often started when someone new joined in, or when going to another location. The interviewer stressed that participants could express themselves in a way they felt comfortable, and that all information shared during the interview was confidential. After having constructed their previous day, participants were given the opportunity to review their previous day again, and add, delete or alter activities if necessary.

*Conflict mapping*

Next, possible conflicts that arose that day were assessed. Although measures focusing on goal *inter-relations and goal interference* are existent, none of them focus on the assessment of *goal conflict* in humans. Our definition of goal conflict was informed by the theoretical accounts of goal conflict by *Lewin (1935)* and *Miller (1944)*. In these accounts, goal conflict is defined as a situation in which the pursuit of one activity or goal competes with the attainment of another, equally valued goal, and which creates at least a temporary stalemate, characterized by an indecisiveness and hesitancy before deciding which activity to pursue (*Miller, 1944*). Patients were provided a definition of goal conflict, and further examples and information. The instructions regarding goal conflict were iteratively developed in collaboration with a group of (pain research) experts and were extensively piloted with patients.

The information provided to the participants about goal conflict was the following. "*Goal conflict is defined as the experience of indecisiveness or doubt about which of two activities to pursue. Examples of conflicts are having doubts whether 'to study for an exam' or 'going out for drinks', 'reading a newspaper' or 'repairing a leaky faucet', or 'resting to reduce pain' or 'going for dinner with friends'. This definition does not incorporate 'social conflict', which is having a fight or an argument.*"

In order to ensure comprehensibility, participants were asked to provide an example that fitted the definition above. Further clarification was given if needed. Participants were then asked to report the conflicts experienced during the previous day. Further information concerning these conflicts was obtained, such as the type of activities involved, the context, reasons of conflict, duration, and decision.

Thirty-one out of 40 (77.5%) patients and 32 out of 37 (86.49%) controls reported at least one conflict. Nine out of 40 (22.5%) patients and five out of 37 (13.51%) controls did not report any conflicts.

*Goal categorization*

After having reported all conflicts, these conflicts were examined more closely. Participants were asked to classify the goal underlying each activity of goal conflicts using the following goal category system (*Chulef, Read & Walsh, 2001*):

1) *Interpersonal/Social*: the goal is to maintain or improve contact or relationships with other people (e.g., going out with friends);

2) *Intrapersonal*: the goal is to maintain or improve personal qualities or personal growth (e.g., be helpful);

3) *Work/Education*: the goal is related to work and/or educational purposes, and is aimed at the personal (academic) career (e.g., following classes, meeting deadlines);

4) *Household*: the goal is to pursue household activities or chores, and is aimed at maintaining or improving your household (e.g., having a clean house);

5) *Leisure*: the goal is to relax or to enjoy yourself, mostly the goal is to pursue activities that are aimed at things you do in your spare time (e.g., hobbies);

6) *Financial*: the goal is to maintain or improve your financial status, freedom, independence, security or stability;

7) *General physical and mental health*: the goal is to maintain or improve your general physical and/or mental health, for example, eating healthy food, stress reduction; with the exception of the goal to avoid, reduce or control pain;

8) *Pain control, avoidance and/or reduction*: the goal is to control, avoid or reduce pain, for example, resting, avoiding movements, taking medication; and

9) *Other*: if the goal does not fit in one of the other categories, this category can be selected.

Participants were informed that only one goal per activity could be selected. If multiple categories were fitting, participants should select the most important one. The list of the goal categories was placed in front of the participant as a reminder. The interviewer also illustrated how to classify the goals of the activities using an example:

*"Imagine sitting in a restaurant and doubting between staying for a chat with your friend, or going back to work. You may want to chat with your friend because you want to invest in the relationship with your friend. This can be placed in the category 'social/interpersonal'. You may want to go back to work because you wish to do the work you are meant to do; this can be classified in the category 'work/education'. However, it is also possible that you wish to go back to work because you want to be a professional and hard-working person, which can be classified in the category 'intrapersonal'. Another goal you may have, is to obtain a financial bonus; this can be placed in the category 'financial'. Since multiple goals are present, you have to pick the one that was most applicable in that situation, for example, 'work'."*

Next, participants themselves classified each activity of the conflicts. This classification allows to identify the type of goal conflict; for example: pain (control/avoidance/reduction) vs. financial. For the purposes of this study, we will refer to a pain-related goal conflict if a pain avoidance/control/reduction is identified as one of the underlying goals in a goal conflict.

*Conflict assessment*

After the goal classification of each conflict, participants were asked to assess a maximum of three conflicts. In case more than three conflicts were reported, the conflicts were selected at random (using a randomization table). As there were two patients reporting more than three conflicts and four controls reporting more than three conflicts, there was no data collection for two conflicts in patients and eight conflicts in controls.

Questions regarding goal conflict involved conflict strength *("How strongly did you experience this conflict?")*, worry *("To what extent did you worry during this conflict?")*, pain-related worry *("To what extent did you worry about pain during this conflict?")*, stress *("To what extent did you feel stressed during this conflict?")*, need of support *("To what extent did you need support during this conflict?")*, conflict solution *("How difficult was it to solve this conflict?")* and solution satisfaction *("How satisfied were you with the solution of this conflict?")*.

Participants also rated the affect during the conflict (11 items, e.g., happiness, sadness, relaxation, frustration). All questions were assessed on a seven-point scale going (0 = not at all; 6 = very much). We ran a principal component analysis on these 11 affect-items. The scree plot analysis revealed two factors with an eigenvalue greater than 1 explaining 74.24% of the variance. The factors created as a result of the factor analysis were (1) *positive affect*, which comprises the variables happy, enthusiastic, and relaxed; and (2) *negative affect*, which comprises the variables sad, nervous, irritated, angry, afraid, powerless, frustrated, and helpless.

## RESULTS

Statistical analyses were performed using SPSS 23.0 and Microsoft® Excel 2010. Alpha was set at 0.05.

The key questions addressed in this paper are:

1) Do patients experience more goal conflict than healthy participants?
2) Do patient and healthy participants differ in the type of conflict experienced?
3) Which goals are most commonly conflicting with pain related goals?
4) Do patient and healthy participants differ in the experience and context of conflict?
5) Can core constructs of the fear-avoidance model or individual differences predict the number of (pain-related) goal conflicts?

### Do patients experience more goal conflict than healthy participants?

The primary objective of this study was to determine the presence of goal conflict in a patient sample and in controls, and investigate whether both groups differ in the frequency of goal conflicts. For this comparison Mann–Whitney $U$ tests were used because the assumption of normality was violated. Patients on average reported 1.53 ± 1.13 goal conflicts (range: 0–4). The total number of conflicts reported by patient participants was 61. Nine patients did not report any conflicts. Control participants reported on average 1.87 ± 1.46 goal conflicts (range: 0–7). Five controls did not report any conflicts. The total

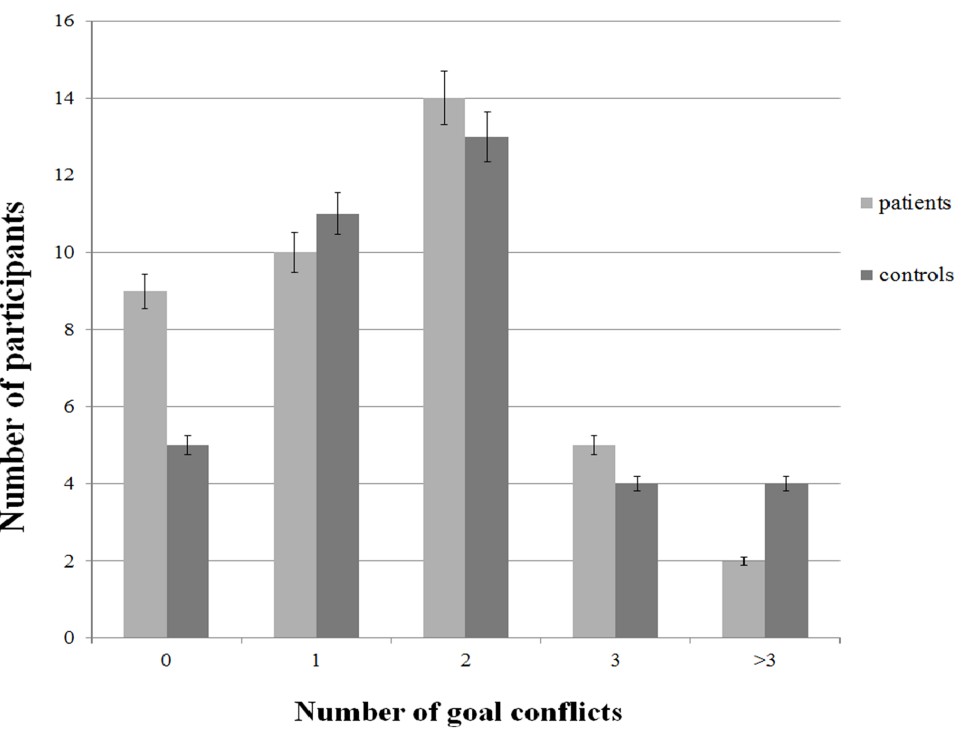

**Figure 1 Frequency of reported goal conflicts as a function of group.**

number of conflicts reported by control participants was 69. There was no significant difference in the number of conflicts between patients and controls ($U = 665.5$, $p = 0.431$). Figure 1 presents the number of participants reporting either no, 1, 2, 3, or more than three goals as a function of group.

## Do patient and healthy participants differ in the type of conflict experienced?

Another aim was to explore whether patients and controls differ in the type of conflicts experienced. More specifically, a motivational account of the fear-avoidance model posits that pain-avoidance goals may compete with other goals in patients with chronic pain. Therefore, we expected that patients experience more pain-related goal conflict than control participants. We assessed whether patients report certain types of conflict more often than control participants. For this purpose, we calculated the number of times that a goal category was used during the goal classification of the conflicts. This resulted in a number of endorsements for each of the nine goal categories per participant.

Mann–Whitney $U$ tests were reported because the assumption of normality was violated. Our tests revealed that on average, patients with fibromyalgia reported more pain-related goal conflicts than control participants, $0.875 \pm 0.991$, and $0.054 \pm 0.229$, respectively, $U = 363$, $p \leq 0.001$. As shown in Table 1, 55% of the patients report at least one pain-related goal conflict whereas only 5.4% of controls did. Furthermore, patients with fibromyalgia on average reported less work-related goal conflicts, $U = 363$, $p \leq 0.001$,

**Table 1** Frequency and percentage of participants reporting pain-related goal conflict.

| Number of pain-related conflicts | Total (N = 77) | | Patients (N = 40) | | Controls (N = 37) | |
|---|---|---|---|---|---|---|
| | N | % | N | % | N | % |
| 0 | 53 | 68.8 | 18 | 45 | 35 | 94.6 |
| 1 | 14 | 18.2 | 12 | 30 | 2 | 5.4 |
| 2 | 8 | 10.4 | 8 | 20 | 0 | 0 |
| 3 | 1 | 1.3 | 1 | 2.5 | 0 | 0 |
| >3 | 1 | 1.3 | 1 | 2.5 | 0 | 0 |

less social-related goal conflicts, $U = 534.5$, $p = 0.021$, and less pleasure-related goal conflicts, $U = 499.5$, $p = 0.004$. Patient and control participants did not differ in the average number of health-related, finance-related, household-related, and intrapersonal-related goal conflicts, $p$s $> 0.05$.

## Which goals are most commonly conflicting with pain-related goals?

Subsequently, we identified the type of goal that participants reported to conflict with the pain-related goal (goal of pain avoidance, control and/or reduction). As mentioned above, patient and control participants reported 61 and 69 goal conflicts, respectively. Of the 61 goal conflicts reported by patients, 35 (57.4%) goal conflicts involved a pain-related goal, whereas only two out of 69 (2.9%) goal conflicts reported by control participants involved a pain-related goal. For patients, the pain-related goal most often conflicted with household goals (45.7%), social goals (20%), and intrapersonal goals (14.3%). Furthermore, pain-related goals conflicted with other health-related goals in 8.6% and with financial goals in 5.7% of reported conflicts. For controls, the two pain-related goal conflicts involved pleasure goals and household goals, respectively.

## Do patient and healthy participants differ in the experience and context of conflict?

As contextual factors might play an important role in the experience of conflict, we compared the contexts between conflicts reported by patients and conflicts reported by healthy controls. Although we did not find any differences in terms of the number of goal conflicts, we expected that patients might experience conflicts as more aversive, and might experience more difficulties in resolving their conflicts. Because the analyses on the experience of conflict were conducted on the conflict level, only participants that reported a conflict, could be included. The analyses were thus run on 61 conflicts reported by 32 controls and 59 conflicts reported by 31 patients.

The context of a conflict pertains to with whom the subject was with during the conflict, where the participant was (location), whether another person caused the conflict, and how the conflict was solved. The frequency and percentage of participants per group is described in Table 2. A conflict of a patient was experienced most often when (s)he was alone (49.2%) or with their family/partner (44.3%). Controls were also most often alone (55%) when experiencing a conflict. The majority of conflicts reported by patients

**Table 2 Frequency and percentage of conflicts per group for the variables who, location, cause, and conflict solution.**

| | Total | | Patients | | Controls | |
|---|---|---|---|---|---|---|
| | *N* | *%* | *N* | *%* | *N* | *%* |
| **Who** | | | | | | |
| Alone | 68 | 52.3 | 30 | 49.2 | 38 | 55.1 |
| Family/partner | 45 | 34.6 | 27 | 44.3 | 18 | 26.1 |
| Friends/acquaintances | 4 | 3.1 | 0 | 0 | 4 | 5.8 |
| Colleagues/fellow students | 5 | 3.8 | 0 | 0 | 5 | 7.2 |
| Other | 4 | 3.1 | 2 | 3.3 | 2 | 2.9 |
| Multiple categories | 4 | 3.1 | 2 | 3.3 | 2 | 2.9 |
| **Location** | | | | | | |
| At home | 93 | 71.5 | 53 | 86.9 | 40 | 58 |
| On the way | 10 | 7.7 | 3 | 4.9 | 7 | 10.1 |
| Visiting family/friends/acquaintances | 4 | 3.1 | 0 | 0 | 4 | 5.8 |
| Work/school | 13 | 10 | 1 | 1.6 | 12 | 17.4 |
| Other | 10 | 7.7 | 4 | 6.6 | 6 | 8.7 |
| **Conflict caused by someone else** | | | | | | |
| No | 98 | 75.4 | 44 | 72.1 | 54 | 78.3 |
| Yes | 32 | 24.6 | 17 | 27.9 | 15 | 21.7 |
| **Conflict solution** | | | | | | |
| Perform 1 of both activities | 85 | 65.4 | 41 | 67.2 | 44 | 63.8 |
| Do both activities (sequentially) | 45 | 34.6 | 20 | 32.8 | 25 | 36.2 |

**Note:**
N, number of participants; %, percentage of participants (within group).

occurred at home (86%), whereas this is less the case for conflicts reported by control participants (58%). School or work accounts for 17.4% of conflicts reported by control participants. For both groups, the conflict was not initiated by others, and the conflict was resolved by doing only one of the activities involved in the conflict.

In order to investigate whether patients and controls differ in the experience of conflict, and to what extent the experience of conflict varies as a function of the number of conflicts we conducted multilevel analysis (on conflicts nested within persons). More specifically, different multilevel analyses are used to explain different measures of experience of conflict (i.e., the outcome variable) as a function of the "dummy" variable patient (controls = 0, patients = 1), the number of conflicts ($N_{conflicts}$) and the interaction between these variables. The variables log (conflict duration), conflict strength, satisfaction, difficulty, worry, worry about pain, stress, and the positive and negative affect factors are used as outcome variables in subsequent multilevel analyses. Using $Y_{ij}$ to represent the score of person $i$ on experience-of-conflict measure $Y$ (the outcome variable) for conflict $j$, the multilevel model can be formulated as follows:

$$Y_{ij} = \alpha_i + \beta_p \text{Patient}_i + \beta_{nc} N_{\text{conflicts}i} + \beta_{p \times nc} \text{Patient}_i * N_{\text{conflicts}i} + \varepsilon_{ij}$$

The error term $\varepsilon_{ij}$ is assumed to have a normal distribution with mean 0 and variance $\sigma_\varepsilon^2$. Furthermore, to account for correlation among the responses of the same person, the

model includes a random intercept $\alpha_i$ that is assumed to have a normal distribution with mean $\mu$ and variance $\sigma_\alpha^2$. To enhance the interpretation of the regression coefficients, the number of conflicts was centered using grand mean centering, so that a value of 0 represents an average number of conflicts. Moreover, in each analysis the dependent variable was standardized to have a mean equal to 0 and a standard deviation equal to 1. As a result, the regression coefficient of the patient dummy ($\beta_p$) indicates how many standard deviations the average predicted $Y$-value increases for patients who reported an average number of conflicts compared to controls who reported an average number of conflicts. Furthermore, the regression coefficient of the number of conflicts ($\beta_{nc}$) indicates how many standard deviations the predicted average $Y$-value increases when persons of the control group report one conflict more. In addition, the coefficient of the interaction ($\beta_{p \times nc}$) indicates the additional increase in the predicted average $Y$-value for patients compared to controls if the person reported one conflict more. Finally, as our sample is relatively small and dependent variables are not always normally distributed, standard errors for estimated parameters are calculated using bootstrapping to increase accuracy. The results of the analysis are presented in Table 3.

The estimated coefficient for the patient dummy variable indicated that (for persons who reported an average number of conflicts) patients reported to worry more during conflicts, $\beta_p = 0.304$, SE $= 0.142$, $p = 0.015$, reported to worry more about their pain, $\beta_p = 1.11$, SE $= 0.123$, $p < 0.001$, reported to be more stressed during a conflict, $\beta_p = 0.68$, SE $= 0.134$, $p < 0.001$, felt more strongly that they needed support during conflicts, $\beta_p = 0.574$, SE $= 0.15$, $p < 0.001$, found their conflicts more difficult to solve, $\beta_p = 0.509$, SE $= 0.14$, $p < 0.001$, were less satisfied with how they solved their conflict, $\beta_p = -0.507$, SE $= 0.162$, $p < 0.001$, experienced less positive feelings, $\beta_p = -0.441$, SE $= 0.133$, $p = 0.001$, and more negative feelings during the conflict, $\beta_p = 0.45$, SE $= 0.131$, $p = 0.001$. Furthermore, assuming an average number conflicts was reported, it took patients longer than controls to solve their conflicts, $\beta_p = 0.56$, SE $= 0.138$, $p < 0.001$. This difference between patients and controls increases 0.346 if one conflict more is reported, $\beta_{p \times nc} = 0.346$, SE $= 0.122$, $p = 0.001$. Lastly, assuming an average number of reported conflicts, patients reported to experience their conflicts more strongly than controls, $\beta_p = 0.601$, SE $= 0.137$, $p < 0.001$. Moreover, the size of this effect increases 0.273 if patients reported one conflict more, $\beta_{p \times nc} = 0.273$, SE $= 0.119$, $p = 0.01$. The number of conflicts did not alter the experience of conflict in either of the groups for all other outcome variables.

### Can core constructs of the fear-avoidance model or individual differences predict the number of (pain-related) goal conflicts?

Because the fear-avoidance model proposes that several factors might play a role in the development of pain-related fear, avoidance, and disability (*Vlaeyen & Linton, 2012*; *Vlaeyen, Crombez & Linton, 2009*; *Vlaeyen, Morley & Crombez, 2016*), we explored whether the amount of pain-related goal conflict—reflected by the number of pain-related goal conflicts—could be predicted by individual differences in process outcomes—such

**Table 3 Multilevel regression analyses for experience of conflict outcome variables.**

| Outcome variable | Predictors | | | | | | | | | | | | Variance components | | | |
|---|---|---|---|---|---|---|---|---|---|---|---|---|---|---|---|---|
| | Mean random intercept | | | Patient | | | Number of conflicts | | | Interaction | | | Error variance | | Variance random intercept | |
| | $\mu$ | SE | $p$ | $\beta_p$ | SE | $p$ | $\beta_{nc}$ | SE | $p$ | $\beta_{p \times nc}$ | SE | $p$ | $\sigma_\varepsilon^2$ | $p$ | $\sigma_\alpha^2$ | $p$ |
| Log (duration) | −0.263 | 0.106 | <0.005 | 0.56 | 0.138 | <0.001 | −0.099 | 0.077 | 0.137 | 0.346 | 0.122 | 0.001 | 0.322 | 0.592 | 0.674 | <0.001 |
| Conflict strength | −0.281 | 0.107 | 0.002 | 0.601 | 0.137 | <0.001 | −0.006 | 0.076 | 0.917 | 0.273 | 0.119 | 0.010 | 0.491 | 0.467 | 0.450 | <0.001 |
| Worry | −0.153 | 0.102 | 0.091 | 0.304 | 0.142 | 0.015 | −0.076 | 0.078 | 0.255 | −0.015 | 0.129 | 0.902 | 0.481 | 0.475 | 0.529 | <0.001 |
| Worry about pain | −0.566 | 0.073 | <0.001 | 1.112 | 0.123 | <0.001 | −0.003 | 0.035 | 0.905 | −0.148 | 0.119 | 0.126 | 0.466 | 0.417 | 0.228 | 0.087 |
| Stress | −0.335 | 0.097 | <0.001 | 0.680 | 0.134 | <0.001 | −0.001 | 0.074 | 0.983 | 0.021 | 0.137 | 0.858 | 0.454 | 0.491 | 0.460 | <0.001 |
| Need for support | −0.283 | 0.089 | 0.002 | 0.574 | 0.150 | <0.001 | −0.029 | 0.046 | 0.434 | 0.118 | 0.140 | 0.316 | 0.594 | 0.398 | 0.347 | 0.007 |
| Difficulty to solve | −0.256 | 0.095 | 0.003 | 0.509 | 0.140 | 0.001 | 0.084 | 0.080 | 0.218 | 0.001 | 0.146 | 0.987 | 0.531 | 0.420 | 0.419 | <0.001 |
| Satisfaction with solution | 0.252 | 0.106 | 0.004 | −0.507 | 0.162 | <0.001 | −0.044 | 0.067 | 0.340 | 0.040 | 0.165 | 0.759 | 0.959 | 0.192 | 0 | 1 |
| Positive affect | 0.197 | 0.104 | 0.032 | −0.441 | 0.133 | 0.001 | 0.040 | 0.098 | 0.662 | −0.159 | 0.160 | 0.256 | 0.376 | 0.503 | 0.632 | <0.001 |
| Negative affect | −0.215 | 0.090 | 0.013 | 0.450 | 0.131 | 0.001 | −0.030 | 0.058 | 0.549 | 0.194 | 0.124 | 0.073 | 0.290 | 0.464 | 0.675 | <0.001 |

Note:
SE, standard error, calculated using bootstrapping; $\sigma_\varepsilon^2$, variance of the error term; $\sigma_\alpha^2$, variance of the random intercept.

as pain-related fear, catastrophizing, and hypervigilance—individual states and traits, such as general anxiety, and individual differences in disability and pain.

Poisson regressions were carried out to assess whether individual differences predicted the number of *pain-related* goal conflicts. Because only two control participants reported a pain-related goal conflict, regressions were carried out with the patient group only ($N = 40$). Measures assessing traits/states included were: positive and negative affect (PANAS), trait anxiety (STAI), Depression, anxiety and stress (DASS), pain catastrophizing (PCS), pain disability (PDI), hypervigilance (PVAQ), pain-related fear (PASS), and cognitive intrusions (ECIP). We also assessed individual differences in disability, years of pain onset, average pain (in a week), pain intensity, and hindrance by pain. We corrected for over- or under-dispersion using a quasi-Poisson approach. Our results indicated that the average number of pain-related goal conflicts reported by patients increased 39.6% for each increase of one standard deviation in average pain, $\beta = 0.396$ (95% CI [0.013–0.778]), Wald $\chi^2 = 4.11$, $df = 1$, $p = 0.043$, 4.3% for every standard deviation increase in anxiety (DASS), $\beta = 0.043$ (95% CI [0.002–0.082]), Wald $\chi^2 = 4.28$, $df = 1$, $p = 0.039$, and 2.5% for each increase of one standard deviation on cognitive intrusions, $\beta = 0.025$ (95% CI [0.006–0.043]), Wald $\chi^2 = 7.011$, $df = 1$, $p = 0.008$. A marginally significant increase of 3.3% and 3.1% in the average number of pain-related conflicts reported were found for an increase of one standard deviation in negative affect, $\beta = 0.033$ (95% CI [−0.001–0.067]), Wald $\chi^2 = 3.6$, $df = 1$, $p = 0.058$, and depression, $\beta = 0.031$ (95% CI [−0.002–0.064]), Wald $\chi^2 = 3.29$, $df = 1$, $p = 0.07$, respectively. None of the other individual difference variables predicted the number of pain-related goal conflicts: pain catastrophizing: $\beta = 0.018$ (95% CI [−0.011–0.047]), Wald $\chi^2 = 1.52$, $df = 1$, $p = 0.218$; positive affect: $\beta = −0.025$ (95% CI [−0.078–0.029]), Wald $\chi^2 < 1$, $df = 1$,

$p = 0.365$; trait anxiety: $\beta = 0.017$ (95% CI [−0.013–0.048]), Wald $\chi^2 = 1.21$, d$f = 1$, $p = 0.272$; stress (DASS): $\beta = 0.023$ (95% CI [−0.011–0.056]), Wald $\chi^2 = 1.79$, d$f = 1$, $p = 0.181$; pain disability: $\beta = 0.02$ (95% CI [−0.011–0.051]), Wald $\chi^2 = 1.56$, d$f = 1$, $p = 0.212$; hypervigilance: $\beta = 0.022$ (95% CI [−0.006–0.050]), Wald $\chi^2 = 2.35$, d$f = 1$, $p = 0.125$; pain-related fear: $\beta = 0.010$ (95% CI [−0.002–0.023]), Wald $\chi^2 = 2.72$, d$f = 1$, $p = 0.099$; disability: $\beta = −0.093$ (95% CI [−0.835–0.649]), Wald $\chi^2 < 1$, d$f = 1$, $p = 0.806$; years of pain onset: $\beta = −0.017$ (95% CI [−0.050–0.017]), Wald $\chi^2 < 1$, d$f = 1$, $p = 0.323$; pain intensity: $\beta = 0.186$ (95% CI [−0.204–0.576]), Wald $\chi^2 < 1$, d$f = 1$, $p = 0.351$; hindrance by pain: $\beta = 0.244$ (95% CI [−0.082–0.530]), Wald $\chi^2 = 2.06$, d$f = 1$, $p = 0.151$.

## DISCUSSION

This study investigated the presence and experience of goal conflicts in patients with fibromyalgia in comparison to healthy controls. For this purpose, 40 patients with fibromyalgia and 37 healthy participants completed a semi-structured interview in which they identified experienced goal conflicts, assessed the experience of the conflict, classified each of their goals in pre-defined categories, and assessed their previous day.

First, we expected patients with fibromyalgia to report more goal conflict than control participants. Both patient and control participants were readily able to report and identify goal conflict. When asked for an example, participants spontaneously reported on personal experiences. These examples often included recurring experiences—patients with fibromyalgia mostly describing conflicts between resting in order to control/reduce pain and doing household chores or going out with friends/family—or examples of great value to the participant (e.g., being able to watch over the grandchildren daily or creating artworks out of ceramic). Nevertheless, our results revealed that patients with fibromyalgia did not spontaneously report more goal conflicts than healthy controls. This finding is not in line with the finding of *Karoly et al. (2008)*. Second, we expected pain patients and controls to differ in the type of conflicts they experience. More specifically, we expected that patients' goal conflicts would include pain avoidance and control more often than those of controls. Indeed, we observed that patients reported more pain-related conflicts than controls. Additionally, patients also reported less conflicts related to work, social, or pleasure goals. Of all conflicts reported by patients, 57.4% involved a pain-goal. Pain goals most often conflicted with household goals (45.7%), social goals (20%) and intrapersonal goals (14.3%). These differences in type of conflict as well as the goals conflicting with pain goals might be due to contextual characteristics, as the participants in our study were mostly women, unemployed and/or receiving disability benefits. For example, patients reporting less work related goal conflict is possibly due to the fact that the majority of patients are unemployed. Another possibility is that patients with fibromyalgia construct their environment in such a way, that they experience as little conflict as possible; or that a recall bias is present, maybe resulting in reporting conflicts pertaining to life domains important to the individual. Therefore, as patients' lives may be predominantly focused on pain, they may experience (and report) less conflict in other domains. Our study is one of the first to reveal the presence of pain-related goal conflicts, and provides preliminary evidence that pain goals conflict with other goals in the

daily life of patients. As such, the inclusion of a broad motivational perspective in the fear-avoidance model is warranted (*Crombez et al., 2012*; *Vlaeyen & Linton, 2012*; *Vlaeyen, Morley & Crombez, 2016*).

Third, another aim was to study the contextual characteristics and the affective experience of the conflict. Regarding the contextual characteristics, our findings demonstrate that patients experienced most conflicts at home (86%), whereas this is less the case for control participants (58%), who also reported experiencing conflicts at work/school or when on their way; again, this may be due to the low employment rate and disability benefits of our patient sample. Both groups reported that they most often experienced a conflict when they were alone. Furthermore, despite the absence of a difference in the number of conflicts they report, patients and controls differed in how they perceive conflict. Overall, it seems that patients experienced conflicts more negatively than controls: they reported less positive and more negative feelings, worried more, felt more stress, and felt more need for support than controls. Patients also perceived their conflicts as more difficult to solve than control participants, and they reported that it took them longer to solve their conflicts. Lastly, patients were on average less satisfied with how they solved their conflicts than control participants. Interestingly, the number of conflicts a participant experienced had little to no impact on the experience of conflict. Our findings are in line with those of *Hardy, Crofford & Segerstrom (2011)*, who studied the relation between goal conflict and fatigue and pain in a sample of 27 females with fibromyalgia. These women were asked to assess pain, distress, and fatigue in the morning and in the evening, and rated their goals and goal conflict in the evening for five consecutive days. They found that pain increased more from morning to evening on days with higher conflict, and women with more symptoms reported more goal conflict than women with fewer symptoms. Taken together, our findings suggest that goal pursuit, and more specifically, goal pursuit in the face of pain, may deplete resources in an already vulnerable population, which may in turn result in more pain and fatigue, or feeling more hampered by it. However, further scientific inquiry is needed to explicitly test these relationships.

The last aim of the current study was to investigate whether individual differences in disability, pain, and core constructs of the fear-avoidance model could predict differences in the amount of pain-related goal conflict. First, we found that higher average pain intensity was associated with a strong increase in the reported number of pain-related conflicts of patients. As these results are correlational in nature, this might indicate that experiencing intense pain may lead to more goal conflict, or conversely, that conflict leads to an increase in pain (*Hardy, Crofford & Segerstrom, 2011*). The relation between pain intensity and the experience of goal conflict warrant further scrutiny. Second, we found that the number of pain-related goal conflicts was associated with a higher number of cognitive intrusions (*Attridge et al., 2015*) as well as more anxiety (*Antony et al., 1998*; *De Beurs et al., 2001*; *Lovibond & Lovibond, 1995a*). Given the importance of pain-related fear and catastrophizing in the fear-avoidance model, we also expected that the greater pain-related fear, and the more catastrophizing, the more conflicts patients would experience. However, our study was unable to demonstrate an impact of pain-related

fear, pain catastrophizing, pain disability, or vigilance. The DRM resulted in a large database. We have only focused on the effects of the frequency (number) of conflicts. Other analyses are also possible. For example, it may be that these constructs not necessarily predict the *number* of pain-related conflicts, but the *characteristics* of the experienced conflict. Further research is needed to investigate this hypothesis. Also of importance is that the number of outcome variables is rather large, and that they might be (strongly) related to each other. It might therefore be useful to investigate which variables are closely related and reliably reflect the impact of goal conflict. Nonetheless, our results demonstrate that expanding the fear-avoidance model with a broad motivational perspective may be fruitful. Our findings indicate that goal conflict or competition in chronic pain is related to the interpretation of a situation as catastrophic, fueled by cognitive intrusions and anxiety. Another intriguing question is whether the characteristics of pain-related conflicts differ from the characteristics of non-pain-related conflicts. This question requires an analysis of the type of goal conflict within subjects. Unfortunately, this analysis was not feasible, because only a limited number of pain and non-pain related goal conflicts was reported, resulting in insufficient power to conduct those analyses on the current dataset.

This study may have clinical implications. The results underscore the importance of the inclusion of goal dynamics in our understanding of chronic pain problems (*Crombez et al., 2012*; *Vlaeyen & Linton, 2012*; *Vlaeyen, Crombez & Linton, 2009*), and provide evidence for the use of treatments focusing on idiosyncratic goal pursuit in other domains aside from pain control and avoidance to improve patients overall well-being and increase physical activity (e.g., motivational interviewing; *Ang et al., 2007*; *Jensen, Nielson & Kerns, 2003*, self-control improvement; *Inzlicht, Schmeichel & Macrae, 2014*). In this paper, we focused on the presence and experience of goal conflicts in a patient sample. Therefore, we only reported if participants pursued none, only one or both goals, but not which specific goal was pursued. Future research might want to assess to what extent patients pursue pain avoidance at the expense of other goals. Our own experience while conducting the interviews suggests that pain avoidance often prevails over other activities, although this was not always the case. Therefore, we suggest that future research investigates whether patients focus on one strategy—that is, prioritizing pain avoidance over other activities—when repeatedly being confronted with a particular type of goal conflict.

Additionally, it might be appropriate to screen for certain individual characteristics such as general anxiety, as these individuals might benefit more from a tailored treatment strategy, since our research suggested that these individuals might experience more pain-related goal conflicts. However, more insight is needed on which patients experience more goal interference than others, or for which patients pain-related goal conflicts weighs more on their physical and psychological well-being.

Some limitations should be considered. First, we had a cross-sectional study design, and no cause-effect relationships can be discerned. Therefore, caution is warranted when interpreting the results. Second, this study is one of the first of its kind, and largely exploratory in nature. Further research is needed to replicate and extend our findings. Third, the DRM generated a large database. To assess the impact of personal characteristics (e.g., fear of pain),

we focused on predicting the number of conflicts. However, other analyses are also possible, and we encourage the use of our database for secondary analyses. Also, a large number of (outcome) variables was collected, which may be dependent. This should be taken into account when looking at the different analyses reported here, or when performing secondary analyses. Fourth, our study sample was limited to patients with fibromyalgia. Therefore, we need to be careful in generalizing our findings to other pain syndromes.

## CONCLUSIONS

This study provides more insight in the dynamic relations between pain-related and other goals and their impact on daily life. At the same time they provide a good starting point to further study the impact of pain-related goal conflict in patients with chronic pain. It seems that goals competing for resources differ between patients and controls, with a more prominent role for pain-avoidance and -control in the lives of patients. Furthermore, our results suggest that patients experience conflict more aversively than healthy controls. However, further scientific inquiry is required to uncover the potential detrimental impact of pain-related goal conflict on daily life experience.

## ACKNOWLEDGEMENTS

The authors thank Eveline Demeulemeester, Julie Mengé, and Nele Decoene for their help in data acquisition and Annick De Paepe for her collaboration. We would also like to thank the Multidisciplinary Pain Centre of Ghent University Hospital, especially prof. Dr. Jacques Devulder, for their help with the recruitment of patients with fibromyalgia. The data of this paper was partially presented during the 10th Congress of the European Pain Federation EFIC, September 6–9, 2017 in Copenhagen, Denmark.

### Funding

This study was supported by the research grant "Pain-Related Fear in Context: The Effects of Concomitant Non-pain Goals and Goal Conflicts on Fear Responding in the Context of Pain" funded by the Research Foundation–Flanders (Fonds Wetenschappelijk Onderzoek [FWO] Vlaanderen), Belgium, granted to Geert Crombez and Johan Vlaeyen (grant ID: G091812N). There was no additional external funding received for this study. The funders had no role in study design, data collection and analysis, decision to publish, or preparation of the manuscript.

### Grant Disclosures

The following grant information was disclosed by the authors:
Pain-Related Fear in Context: The Effects of Concomitant Non-pain Goals and Goal Conflicts on Fear Responding in the Context of Pain.
Research Foundation–Flanders (Fonds Wetenschappelijk Onderzoek [FWO] Vlaanderen), Belgium, granted to Geert Crombez and Johan Vlaeyen: G091812N.

## Competing Interests

The authors declare that they have no competing interests.

## Author Contributions

- Nathalie Claes conceived and designed the experiments, performed the experiments, analyzed the data, contributed reagents/materials/analysis tools, prepared figures and/or tables, authored or reviewed drafts of the paper, approved the final draft.
- Johan W.S. Vlaeyen conceived and designed the experiments, contributed reagents/materials/analysis tools, authored or reviewed drafts of the paper, approved the final draft.
- Emelien Lauwerier conceived and designed the experiments, contributed reagents/materials/analysis tools, authored or reviewed drafts of the paper, approved the final draft.
- Michel Meulders analyzed the data, contributed reagents/materials/analysis tools, authored or reviewed drafts of the paper, approved the final draft.
- Geert Crombez conceived and designed the experiments, contributed reagents/materials/analysis tools, authored or reviewed drafts of the paper, approved the final draft.

## Human Ethics

The following information was supplied relating to ethical approvals (i.e., approving body and any reference numbers):

The current study was approved by the Medical Ethical Committee of Ghent University Hospital (project 2014/0667—Belgian registration number: B670201421583).

## Data Availability

The raw data are provided in the Supplemental Files.

## Supplemental Information

Supplemental information for this article can be found online at http://dx.doi.org/10.7717/peerj.5272#supplemental-information.

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
