# Peer review of "Goal conflict in chronic pain: day reconstruction method"

_PeerJ, doi:10.7717/peerj.5272_

## Round 0.1 · original submission · Minor Revisions

Please revise the manuscript accordingly to the reviewer comments; especially focusing on the comments of reviewer 1. looking forward to reading the revised manuscript soon!

Reviewer 1 ·

Basic reporting

The manuscript is generally well prepared. However, it does contain numerous errors such as number of participants not being equal to numbers approached minus those who refused! In addition more work could be done to make it clearer why all the measures were used and to what purpose.

Experimental design

The design was a matched control group which seems appropriate. However, it is not clear how well matched the groups are on variables other than sex, age and education. Are they matched on paid employment, partner etc.

Otherwise the methods were well described. Although it struck me as off that the conflicts measure did not seem to examine what the conflicts were between (e.g., work vs relationships, pain reduction vs exercise). It was unclear to me how to classify a conflict that was between a goal in one goal category and a goal in a distinct category. This needs to be made clearer.

Validity of the findings

The results are generally well presented and clear.

A report of which of the nine categories the goal conflicts were classified into would be useful.

It was also not clear why number of conflicts was focused on to be predicted by other variables in Table 3 or how this analysis was conducted. Would it not be more interesting in the current context to examine predictors of the strength of the conflict?

Additional comments

This work explores some interesting issues although the value of the work is obscured by the way it is presented. The authors could do more to highlight what are the key questions here and link these to specific analyses. At present the manuscript is unclear and sometime confusing. A revision should try to present a clearer, focuses manuscript and perhaps remove some of the less central analyses.

·

Basic reporting

The manuscript is unique because there is no previous, such detailed info about goal conflicts in chronic pain assessed with daily reconstruction method as well as there is lack of scientific publications about patients suffering from chronic pain and their daily conflicts.The introduction unit is very consistent and precisely describes the research.The manuscript does not need to be proofread by native speaker however inside abstract passive voice should be used more often than descriptive language as well as Authors should decide if they use numbers or words.There are about 57 cited publications including earlier ones and the latest. Most of the references are with impact factor.The results are clearly presented with figures and tables. The Manuscript is written according to the scientific manner with all needed structure inside.

Experimental design

The manuscript is original because it fills the gap of knowledge about understanding of goal conflicts in patient with chronic pain. Research tasks are very well defined and clearly proceeded and all the results are meaningful.Experimental investigation is performed to a high technical and ethical standards.The methodology is fully compatible with good scientific practice.The discussion is logical and rich in details. All the obtained results were described in details and discussed with available literature.

Validity of the findings

The manuscript is the novel publication about goal conflict in chronic pain. It is the very correct and reliable research manuscript describes the types of daily conflicts with the detailed analysis of its reasons and possible effects. All used data are harvest with the highest ethics and are analyzed in the very approved manner. Conclusions are well stated and linked directly to original research.

Additional comments

Dear Authors.Great job and I wish You all the further success and publications.

---

## Round 0.2 · Minor Revisions

Overall, the quality of the manuscript has been improved significantly!
Please address the minor revisions made by reviewer 1 and submit a revised manuscript. Looking forward to seeing this.

Reviewer 1 ·

Basic reporting

n/a

Experimental design

n/a

Validity of the findings

n/a

Additional comments

PeerJ #23277

The authors have generally done a good job of responding to reviewer’s comments and improving the manuscript. However, a few issues remain.

1. How were the 55 out of 181 control participants selected?
2. Section 3.4. Why report the data on numbers who did not report conflicts here rather than earlier before any data on conflicts are reported? It would be useful to report n out of N alongside each percentage reported. The description of the multi-level models was still unclear and was only understandable by consulting Table 3. In addition, it seems problematic to look at each of these models as independent when the long list of dependent variables are probably strongly related (or at least not independent). Perhaps this could be noted in the discussion.
3. In the discussion I thought the authors could do more to note inter-dependencies. Presumably if you come up with more conflicts about pain you probably mention less conflicts about other issues (given the number of conflicts people report is similar)? Also how meaningful/useful is it to note that patients mention more conflicts at home and less at work if they are less likely to be in work!

·

Basic reporting

The Manuscript is unique because there is no previous, such detailed info about goal conflicts in chronic pain assessed with daily reconstruction method as well as there is lack of scientific publications about patients suffering from chronic pain and their daily conflicts. The introduction unit is very consistent and precisely describes the research. The Manuscript was read by the native speaker that is why reading, understanding and text flow were improved. There are 57 cited publications including earlier ones and the latest. Most of the references come from impact factor journals. The Results are clearly presented and illustrated with tables and figures. The Manuscript is written according to the scientific manner with all needed structure inside.

Experimental design

The Manuscript is original because it fills the gap of knowledge about the understanding of goal conflicts in patients with chronic pain. Research tasks are very well defined and clearly proceeded and all the results are meaningfull. The experimental investigation is performed to a high technical and ethical standard. The methodology is fully compatible with good scientific practice. The discussion is logical and rich in details. All the obtained results were described in details and discussed wit available literature.

Validity of the findings

The Manuscript is the novel publication about goal conflict in chronic pain. It is very correct and reliable research manuscript and describes the types of daily conflicts with the detailed analysis of its reasons and possible effects. All used data are harvest with the highest ethics and are analyzed in the very approved manner. Conclusions are well stated and linked directly to original research.

Additional comments

After corrections it is even better and easier to read and follow. The whole text is very consistent. I wish all the team all the further successes.

---

## Round 0.3 · accepted · Accept

The current version of the manuscript meets all criteria for publication.

#